# Autophagy is important to the acidogenic metabolism of *Aspergillus niger*

**Baljinder Kaur, Narayan S. Punekar** *

Metabolism and Enzymology Laboratory, Department of Biosciences and Bioengineering, Indian Institute of Technology Bombay, Mumbai, India

* nsp@iitb.ac.in

## Abstract

Significant phenotypic overlaps exist between autophagy and acidogenesis in *Aspergillus niger*. The possible role of autophagy in the acidogenic growth and metabolism of this fungus was therefore examined and the movement of cytosolic EGFP to vacuoles served to monitor this phenomenon. An autophagy response to typical as well as a metabolic inhibitor-induced nitrogen starvation was observed in *A. niger* mycelia. The vacuolar re-localization of cytosolic EGFP was not observed upon nitrogen starvation in the *A. niger Δatg1* strain. The acidogenic growth of the fungus consisted of a brief log phase followed by an extended autophagy-like state throughout the idiophase of fermentation. Mycelia in the idiophase were highly vacuolated and EGFP was localized to the vacuoles but no autolysis was observed. Both autophagy and acidogenesis are compromised in *Δatg1* and *Δatg8* strains of *A. niger*. The acidogenic growth of the fungus thus appears to mimic a condition of nutrient limitation and is associated with an extended autophagy-like state. This crucial role of autophagy in acidogenic *A. niger* physiology could be of value in improving citric acid fermentation.

## Introduction

*Aspergillus niger* is a filamentous fungus of industrial and commercial importance. It is capable of growing on simple and inexpensive substrates and secretes large amounts of proteins, metabolites and organic acids [1]. This fungus is industrially famous for citric acid production [2]. The acidogenic growth requires a specific set of media conditions with low pH, nitrogen limitation [3] and Mn deficiency [4] as major features. Despite numerous studies addressing the biochemistry and physiology of this fungus, no single, clear cause-effect relation describing acidogenesis has emerged [5]. Acidogenic *A. niger* mycelia are highly vacuolated and the fraction of vacuolated mycelia increases along with the citrate production [3,6,7]. Many of the biochemical, physiological and morphological features of *A. niger* acidogenic growth mirror those observed during fungal autophagy (Table 1). Although both conidiation and germination are influenced by the organism's state in autophagy and acidogenesis, these effects appear to be divergent. The phenotypic overlaps between autophagy and acidogenesis suggest that the two

**Data Availability Statement:** All relevant data are within the paper and its Supporting Information files.

**Funding:** This work was supported partly by the funds from the Department of Biotechnology, Government of India (http://dbtindia.gov.in/) (grant

no. BT/PR4827/PID/6/647/2012) to NSP. BK
received a University Grants Commission, India
(ugc.ac.in) fellowship (328469) during the course
of her research. The funders had no role in study
design, data collection and analysis, decision to
publish, or preparation of the manuscript.

**Competing interests:** The authors have declared
that no competing interests exist.

processes may be related and that autophagy may play an important role during *A. niger* citric acid fermentation.

Autophagy is a highly conserved process ubiquitous to eukaryotes wherein the cellular constituents are degraded inside their own vacuole/lysosome. *Saccharomyces cerevisiae* has been studied extensively to define autophagy pathways and organelle specific autophagy [17,18]. Autophagy provides an efficient means to recycle and translocate the contents of ageing hyphae for the benefit of the mycelial entity without losing them into the surrounding environment [19–22]. It is important for normal growth of *Aspergilli* as well as their survival during starvation and ageing [8–10,12,23–27]. Autophagy is involved in germination and conidiation of *Aspergillus oryzae* and *Aspergillus fumigatus*; the autophagy mutants show poor conidiation even under non-autophagic conditions and this could be rescued by supplementing the nitrogen source [10]. The cell organelles like peroxisomes, mitochondria and nuclei from the basal cells of *A. oryzae* are subjected to autophagy even during nutrient rich growth conditions [21,27]. The secretion of enzymes and metabolites by filamentous fungi is also influenced by autophagy. Blocking autophagy by disrupting key *atg* genes resulted in a three-fold increase in the production of bovine chymosin by *A. oryzae* [28]. Impaired degradation of peroxisomes and delayed mycelia deterioration led to a 37% increase in penicillin yields in the *atg1* deleted strain of *Penicillium chrysogenum* [29].

Observations like highly vacuolated state of acidogenic mycelia, increased protein degradation coupled to nutrient-limited growth [4,6,13] suggest a link between citric acid fermentation and autophagy in *A. niger*. Whether autophagy occurs and is involved in the acidogenic metabolism of this fungus was therefore examined. The movement of cytosolic EGFP to vacuoles was used as a tool to monitor autophagy [9,25,26] and we show that an autophagy-like state exists throughout the acidogenic stages of fermentation and a block in autophagy pathway negatively impacts citric acid production.

## Materials and methods

### Strains, cultivation conditions and sampling

Various *A. niger* strains expressing EGFP in their cytoplasm were used. These included C6JT2 strain (NCIM 1380) [30] while AR19#1, BN56.2 (*Δatg1*), AW24.2 (BN56.2 complemented with *atg1*), BN57.1 (*Δatg8*) and AW25.1 (BN57.1 complemented with *atg8*) [25] were obtained from Fungal Genetics Stock Center, KS, USA. The fungal strains were maintained on yeast dextrose agar [31]. This medium was either supplemented with 1.0 mg mL$^{-1}$ phosphinothricin (for C6JT2 strain with a *bar* marker) or 100 μg mL$^{-1}$ hygromycin (for strains BN56.2, AW24.2, BN57.1 and AW25.1). Whereas *A. niger* NCIM 565 and AR19#1 (both wild type strains) were cultured on potato dextrose agar [31].

**Table 1. Autophagy and acidogenesis–A comparison.**

| Acidogenesis | Autophagy |
|---|---|
| Acidogenesis requires nitrogen-limited medium [3] | Nitrogen starvation induces autophagy [8–10] |
| Trace metals deficiency [3] | Metal ion starvation induces autophagy [10] |
| Germination is impaired during acidogenic growth [11] | Delayed germination in the absence of autophagy [12] |
| Reduced conidiation during acid production [3] | Reduced conidiation in the absence of autophagy [12] |
| Enhanced intracellular protein degradation [13] | Increased protein degradation during autophagy [14] |
| Accumulation of intracellular amino acids [15] | Autophagy leads to elevated amino acid pool [14] |
| Enhanced vacuolization observed during acidogenesis [6] | Number of vacuoles increases during autophagy [16] |

All the experiments were performed under shake flask conditions (at 30°C and 220 rpm). A total of $10^8$ spores (harvested from suitable solid media in Petri plates) were inoculated into 100 ml of respective liquid media in one liter Erlenmeyer flasks. The fungal strains were cultured either on the minimal medium (MM; which contained 20.0 g/l glucose, 3.0 g/l $KH_2PO_4$, 6.0 g/l $Na_2HPO_4$, 0.5 g/l $MgSO_4.7H_2O$, 2.25 g/l $NH_4NO_3$, 10 mg/l $ZnSO_4.7H_2O$, 3.0 mg/l $MnSO_4.7H_2O$, 1.5 mg/l $Na_2MoO_4.H_2O$, 20.0 mg/l $FeCl_3.6H_2O$ and 1.0 mg/l $CuSO_4.H_2O$. The medium pH was adjusted with 0.1 N HCl to 5.5–6.0) or on the acidogenic medium (AM; which contained 140.0 g/l sucrose, 1.0 g/l $KH_2PO_4$, 0.1 g/l $Fe(NH_4)_2(SO_4)_2$, 2.25 g/l $NH_4NO_3$ and 0.25 g/l $MgSO_4.7H_2O$ for eight days) [31]. Since citric acid fermentation is highly sensitive to presence of trace metals (especially $Mn^{2+}$ ions) all the glassware was treated first with 20% nitric acid and then thoroughly washed with double distilled water. The minimal medium prepared without the addition of $NH_4NO_3$ (MM-N) served to create nitrogen starved condition.

The mycelia harvested from shake flask cultures were harvested by drying between filter papers. The harvested biomass was immediately frozen in liquid nitrogen and stored at −20°C till further use. The spent medium was also frozen and stored similarly for further analysis. For biomass dry weight measurements mycelia were dried in hot air oven (70°C) to constant weight (4–5 days).

### *A. niger* morphology

The *A. niger* was normally grown on MM (200 mL in one L flasks) in shake flasks for 20 h (log phase) and the mycelia were then harvested, washed with cold sterile distilled water and transferred to nitrogen deprived media (MM-N). Parallelly, an equal amount of mycelial biomass (~ 2.0 g wet weight) was transferred back to fresh MM to serve as control. These resuspended mycelia were again incubated for 4 h (at 30°C and 220 rpm) and sampled for microscopy. Washed *A. niger* mycelia were also transferred to MM containing either dimethyl isophthalate (DMIP; 3.0 mM) or citric acid (10 mM or 100 mM, pH adjusted) to test their effects. Stock DMIP solution was prepared in acetone because of its limited solubility in water and suitable solvent controls were included for such experiments.

*A. niger* strains were inoculated ($10^8$ spores per 100 mL culture medium) in either MM or AM media and grown as shake flask cultures (at 30°C and 220 rpm) for 8 days. The samples of mycelia and corresponding spent media were obtained (separate flasks for each time point) for microscopy and citrate analysis, respectively.

### Microscopy and image analysis

Freshly harvested *A. niger* mycelia were directly observed under the microscope without any fixation step. The Olympus fluorescence inverted microscope model (IX83) equipped with plan-apochromat 100×/1.4NA objective lens and DIC was used for capturing fluorescent images. The GFP 4050 B-000 filter (excitation-466/40-25, emission- 525/50-25) used for capturing EGFP fluorescence and DAPI-5060C filter (excitation-377/50-25, emission- 447/60-25) was used for CMAC visualization. The mitotracker and FM4-64 staining was followed using a combination of TRITC-B-000 filter (excitation-543/22-25) and LF561/LP-C-000 (emission-561R), respectively. A Zeiss Axio-Observer Z1 inverted confocal microscope equipped with iplan-apochromat 63×/1.4NA objective lens was used to visualize EGFP fluorescence (488 nm argon laser), FM4-64 and mitotracker stain (561 DPSS laser) and CMAC (multiphoton laser). The images were deconvoluted to enhance signal to noise ratio [32] and analyzed using respective software (CellSense, Zen black or ImageJ).

The culture media of mycelia at different growth stages were replaced with the respective fresh media containing 5 μM FM4-64. After incubation for 10 min (at 30°C and 220 rpm) the

mycelia were quickly harvested, washed twice with respective media without the dye and further incubated for 10 min; these were subjected to microscopy. For vacuolar staining with CMAC, the dye was directly added (at a final concentration of 10 μM) to an aliquot of culture medium. These samples were incubated for 15 min (at 30°C and 220 rpm) before observing under the microscope [33].

Typically, a total of 50 micrograph of each cell type were imaged from five independent experiments. Unless otherwise mentioned, the scale bar in the images corresponds to 2 μm.

### *A. niger* genomic DNA PCR

The frozen *A. niger* mycelia were crushed in liquid nitrogen to form fine powder; this was followed by all the steps as recommended for QIAGEN DNeasy Plant Mini kit for genomic DNA isolation. The genomic DNA was used for characteristic PCR according to standard molecular biology procedures [34]. In all PCR reactions—3.5 mM $MgCl_2$, 250 μM dNTPs, 0.5 μM primers and 5 units of *Pfu* polymerase (MBI Fermentas) were present in a final volume of 100 μL. The PCR products were subjected to electrophoresis on 1% agarose gels. The primers used in study are listed in S1 Table.

The three *A. niger* strains namely, AR19#1 (wild type), BN56.2 (Δ*atg1*) and AW24.2 (*atg1* complemented) were subjected to genotypic characterization, in terms of the presence and/or absence of *atg1*, *atg8* and *hph* (marker used to disrupt *atg1*) genes, before use. Their respective genomic DNA was subjected to diagnostic PCR using the primer pair diagnRP/diagnatg1FP (for *atg1* gene), Atg8FwdP/ATG8cDNARP (for *atg8* gene) and HYnested/YGnested (for *hph* gene). As expected, the characteristic PCR amplification patterns confirmed that the *atg1* sequence was deleted in BN56.2 strain, *hph* gene was absent in AR19#1 strain and that *atg8* gene was intact in all three strains.

### Citric acid and protein estimations

The concentration of citric acid in the spent medium was determined using citrate lyase and phenylhydrazine as reported before [31]. Citrate lyase (Roche Diagnostics India Pvt. Ltd.) from *Aerobacter aerogenes* was used in these estimations. The citrate data on day 8 of acidogenesis for all three strains was statistically evaluated. The one way ANOVA was run on the data for citrate concentration in the spent media as well as on the citrate yield per gram mycelial dry weight.

The frozen fungal mycelia were crushed and protein extracted with buffer [31]. Protein from these cell free extracts was estimated by Bradford's method [35] with crystalline bovine serum albumin as standard.

## Results and discussion

The filamentous growth of *A. niger* comprise variously entangled hyphae to result in the formation of loose or tight pellets and dispersed mycelia [36,37]. The hyphal tubes are differentiated into tip cells, followed by few intermediate cells and inactive basal cells (Fig 1). This heterogenous arrangement of cells makes it interesting yet difficult to study and generalize on the cellular phenomena like autophagy. Autophagy has evolved to recycle and translocate the contents of aging hyphae for the benefit of the mycelial entity without losing them into the surrounding environment [21]. The basal cells or older hyphae are nearly filled with large vacuoles and enter autolysis subsequently [33,38]. Hence the mycelial morphology and subcellular organelles of viable tip cells (and the following few intermediate cells) were monitored to study autophagy in the acidogenic *A. niger*. The pH sensitivity of tagged-EGFP fluorescence in the *A. oryzae* vacuoles *in vivo* is reported [39]. However, the intracellular pH homeostasis of *A. niger*

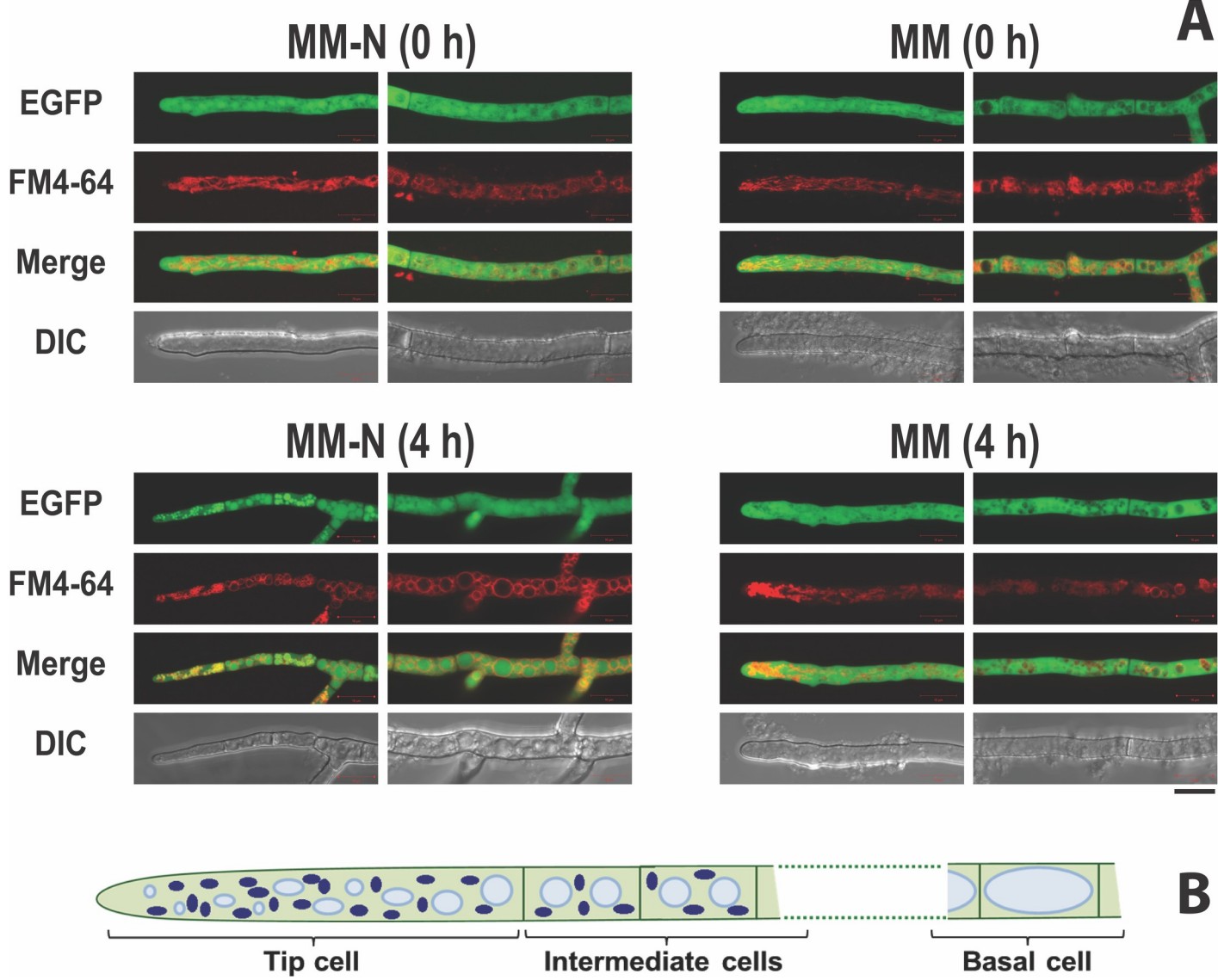

**Fig 1. *A. niger* mycelia respond to nitrogen starvation. (A)** The mycelia of *A. niger* C6JT2 strain grown (up to 20 h) on MM were transferred to MM-N. The EGFP fluorescence of mycelia at the time of transfer (MM-N, 0 h) and after 4 h of N starvation (MM-N, 4 h) was monitored. The corresponding non-starved mycelia control is also shown. In each panel, the left frame shows the image of a tip cell and the right frame is of an intermediate cell. The vacuolar membranes of mycelia were stained red with FM4-64 (scale bar = 10 μm). **(B)** Schematic of *A. niger* hypha. The nuclei (in dark blue), vacuoles (in light blue) and the septa (dark green lines between cells) are shown. The dashed lines represent the connecting intermediate cells up to the basal cell.

during acidogenic growth is well documented. Both the vacuolar pH (of 6.2) and cytoplasmic pH (of 7.6) of the fungus do not vary much when the extra-cellular pH was varied between 2.5 and 9.5 (and also in the presence of citrate) [40,41]. The movement of EGFP from the *A. niger* cytoplasm to its vacuoles could thus be studied independent of the pH effects during fermentation. Relocation of cytosolic EGFP to vacuoles was therefore conveniently exploited and used to mark autophagy in *A. niger* mycelia.

## Nitrogen starvation induces EGFP localization to vacuoles in *A. niger*

Autophagy involves the transport of cytoplasmic components to the lysosome/vacuole for degradation and it is strongly induced by nutrient starvation. The movement of cytosolic EGFP to

vacuoles has served well to monitor autophagy in response to carbon/nitrogen starvation in *S. cerevisiae* [14], *A. oryzae* [8,9], *A. niger* [25], *Aspergillus nidulans* [26] and *P. chrysogenum* [29]. That autophagy occurs with carbon depletion in submerged *A. niger* cultures is well established but the study with respect to nitrogen limitation was restricted to surface cultures on solid MM [25]. Also, the Δ*atg8* strain was shown to be more sensitive to nitrogen limitation than the Δ*atg1* strain, whereas both deletion strains were comparably affected by carbon limitation. Since acidogenesis is possibly a nitrogen limited state of submerged growth, it was of interest to first demonstrate that nitrogen starvation leads to autophagy in two different *A. niger* cultures.

Two *A. niger* strains expressing EGFP in their cytosol [25,30] were used to visualize autophagy upon nitrogen starvation. The log phase mycelia of C6JT2 and AR19#1 strains were transferred to a nitrogen starvation medium (MM-N). Significant EGFP movement to vacuoles could be observed within 2 h after the transfer of mycelia to MM-N. However, this was complete by 4 h and by then the EGFP was localized exclusively in vacuoles, particularly of the intermediate cells (Figs 1 and 2). The vacuoles at this stage were generally more in number and larger in size and were filled with EGFP. The tip cells contained fewer but smaller vacuoles. In the normal (transferred to MM for 4 h) non-starved mycelia EGFP fluorescence remained exclusively outside the vacuoles, both in the tip and the intermediate cells (Fig 1). The results confirmed that nitrogen starvation induces the movement of cytosolic EGFP to vacuoles in *A. niger*. Both the strains behaved similarly when subjected to carbon starvation as well (not shown).

The effect of nitrogen starvation on the biomass yield of four *A. niger* strains (AR19#1, BN56.2, AW24.2 and C6JT2) was also monitored. There was little increase in the biomass after 4 h of transfer to MM or MM-N. But the growth difference was obvious when the mycelia were allowed to grow for 24 h. The fungal biomass increased 3 to 4 fold after transfer on MM while biomass gain on MM-N was marginal (Table 2). On an equal biomass wet weight basis, the total extractable mycelial protein content was substantially lower in the nitrogen starved mycelia (S1 Fig). The 24 h nitrogen starved mycelia generally showed increased vacuolation and cryptic growth in the form of many thinner mycelia was clearly observed (not shown). The cryptic growth and occurrence of thin mycelia (compared to the regular mycelia growing on normal media) are generally observed in starved cultures [42–44].

## Autophagy induced by nitrogen starvation in *A. niger* is *atg1* dependent

The autophagy phenomenon is best established by its impairment in the corresponding *atg* gene disrupted/deleted strains. While our efforts to disrupt *atg1* gene in the background of C6JT2 strain were ongoing, an analysis of autophagy by disrupting *atg* genes (in the background of *A. niger* N402 strain) was reported [25]. The three relevant *A. niger* strains (all expressing EGFP in the cytosol) namely, AR19#1 (wild type), BN56.2 (*atg1* deletion; Δ*atg1*) and AW24.2 (*atg1* complemented) from that study were first used to confirm nitrogen starvation induced autophagy. The log phase grown mycelia of these three strains were subjected to nitrogen starvation (by transfer to MM-N medium; see above) and the mycelia were observed after 4 h. The cytosolic EGFP was localized to vacuoles in AR19#1 strain but remained in the cytosol of *atg1* deleted strain BN56.2 (Fig 2). However, the vacuolar relocation of EGFP was restored upon *atg1* complementation (in AW24.2 strain). The results clearly demonstrate that autophagy is induced when the fungus experiences nitrogen starvation directly or indirectly by nitrogen metabolic inhibition (see below).

## Blocking ammonia assimilation also induces autophagy in *A. niger*

Besides nutrient starvation, autophagy can be induced by treatment with rapamycin, an inhibitor of Tor (target of rapamycin) kinase [23]. Inhibition of nitrogen metabolism, blocking

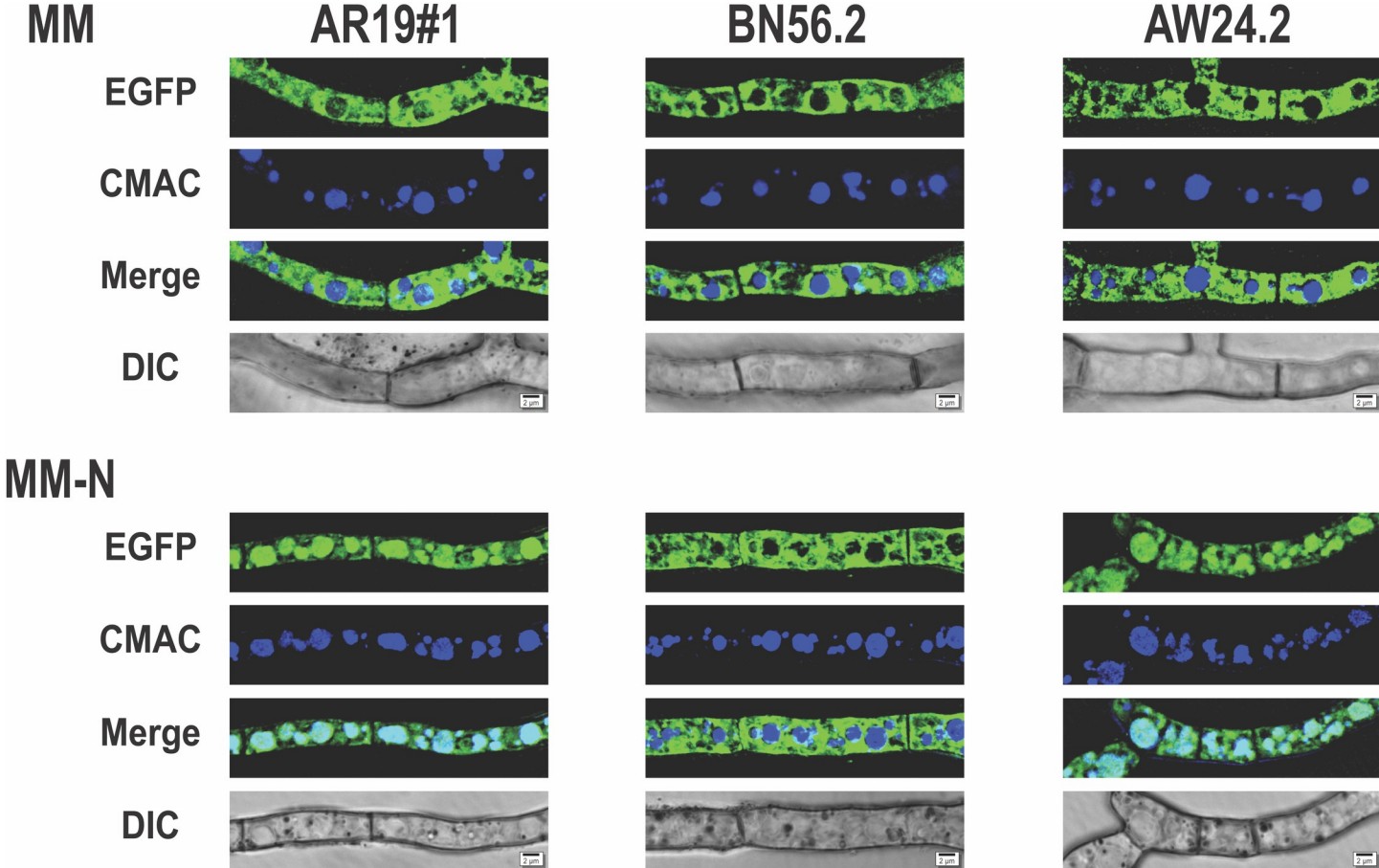

**Fig 2. Autophagy response by nitrogen starved *A. niger* mycelia.** The mycelia of *A. niger* strains AR19#1 (wild type), BN56.2 (Δ*atg1*) and AW24.2 (*atg1* complemented) grown on MM till 20 h were transferred to MM-N. The EGFP fluorescence of mycelia after 4 h of N starvation is shown along with corresponding non-starved mycelia control. Each frame shows the image of an intermediate cell. The mycelia were stained with CMAC when vacuoles appear blue; the vacuoles containing EGFP appear cyan on merge (scale bar = 2 μm).

ammonia assimilation for instance, could also create a nitrogen starvation-like condition. Incubation with dimethyl isophthalate (DMIP) and subsequent inhibition of NADP-glutamate dehydrogenase *in vivo* resulted in growth inhibition and extensive vacuolation of *A. niger* mycelia [45]. Treatment with DMIP should mimic nitrogen starvation even when plenty of nitrogen (as ammonium) is available in the growth medium. This was tested by incubating log phase *A. niger* mycelia with 3 mM DMIP in MM. Autophagy was induced within 3 h in such

**Table 2. Effect of nitrogen starvation on the biomass yield of *A. niger*.**

| *A. niger* strain | Biomass (wet weight, g)[a] | | |
|---|---|---|---|
| | at 0 h | 24 h on MM-N | 24 h on MM |
| AR19#1 (wild type) | 6.5 | 6.9 | 19.5 |
| BN56.2 (Δ*atg1*) | 5.3 | 5.5 | 15.6 |
| AW24.2 (*atg1* complemented) | 5.1 | 6.2 | 15.0 |
| C6JT2 (NCIM 1380) | 2.7 | 3.6 | 11.4 |

[a] Average of three separate experiments

mycelia and the cytosolic EGFP was translocated to vacuoles. Also, the DMIP induced translocation of EGFP to vacuoles was adversely affected in *A. niger* BN56.2 (*atg1* deleted strain) but was restored by *atg1* complementation (in AW24.2 strain) (S2 Fig). The DMIP effect thus offers a unique model to induce autophagy by means of a specific metabolic inhibitor (or an antimetabolite).

### Autophagy is integral to acidogenesis in *A. niger*

It is generally believed that acidogenesis by *A. niger* represents a nutrient limited growth with a major imbalance in C:N ratio of the fermentation medium [3,4]. Perceived nitrogen starvation (see above) may therefore push the fungus towards autophagy. Significant phenotypic overlaps between autophagy and acidogenesis (Table 1) suggested that the two processes may be interconnected. This was directly tested by growing *A. niger* on fermentation medium (AM) and monitoring the mycelial morphology. The normal growth of filamentous fungi (such as the growth of *A. niger* on MM) typically involves distinct stages: the growth phase which later transitions into autolysis of the older mycelia [44]. However, during acidogenic growth *A. niger* goes through two broad phases namely, the trophophase (the growth phase) and the idiophase (citric acid production phase). The fate of cytosolic EGFP was monitored through these two phases of acidogenesis.

The EGFP fluorescence was in the cytoplasm of the AR19#1 strain (wild type) during the trophophase (up to day 1) but in the subsequent idiophase (day 2 to 8) it was localized with the vacuoles (Fig 3). This idiophase was characterized by highly vacuolated mycelia with EGFP exclusively found in the vacuoles. In contrast, the movement of EGFP to vacuoles was not observed in the autophagy impaired *A. niger* BN56.2 strain. Upon *agt1* complementation (in AW24.2 strain) however, the movement of EGFP to vacuoles was restored (Fig 3). Comparable results were obtained when autophagy was manipulated by *agt8* deletion (BN57.1 strain) and its subsequent complementation (AW25.1 strain) (S3 Fig). Clearly, an autophagy-like state of mycelial morphology exists throughout the acidogenic stages (idiophase) of citric acid fermentation. This extended autophagy phase was not followed by autolysis (even after 20 days of fermentation). Whether the observed phenotype could possibly be due to the presence of citrate in the medium was also tested. The mycelial morphology of all three strains remained unaffected even after 8 h of incubation with citrate (added at 10 mM and 100 mM, to log phase *A. niger* mycelia grown in MM), indicating that the presence of extra-cellular citrate did not cause the fungal autophagy response. A systematic study to unravel the role of other factors (listed in Table 1) on acidogenic metabolism, morphology and associated autophagy response is useful/ underway.

The effect of impaired autophagy on citric acid fermentation was as also studied. The three *A. niger* strains namely, AR19#1 (wild type), BN56.2 (Δ*atg1*) and AW24.2 (*atg1* complemented) were grown on AM and citrate levels in the spent media were measured during the course of fermentation. Acidogenesis was compromised in both Δ*atg1* (Fig 4) as well as Δ*atg8* (S4 Fig) strains of *A. niger*. The earliest time point at which citric acid was detected coincided with the movement of EGFP from cytoplasm to vacuoles in the AR19#1 strain (Fig 4A; day 2 onwards). The citric acid production in the autophagy impaired *A. niger* (BN56.2) strain was significantly reduced (p < 0.0001) (Fig 4B); the drop in citrate formation was alleviated significantly by *atg1* complementation (in the AW24.2 strain). A delay in citrate production in AW24.2 (*atg1* complemented) was observed (Fig 4A, top panel). This strain is not strictly comparable to the parent strain AR19#1 for the deletion (in BN56.2 strain) was complemented by random integration of the *atg1* cassette and not by homologous gene replacement [25]. All the three strains showed comparable increase in biomass during their growth on AM as well as

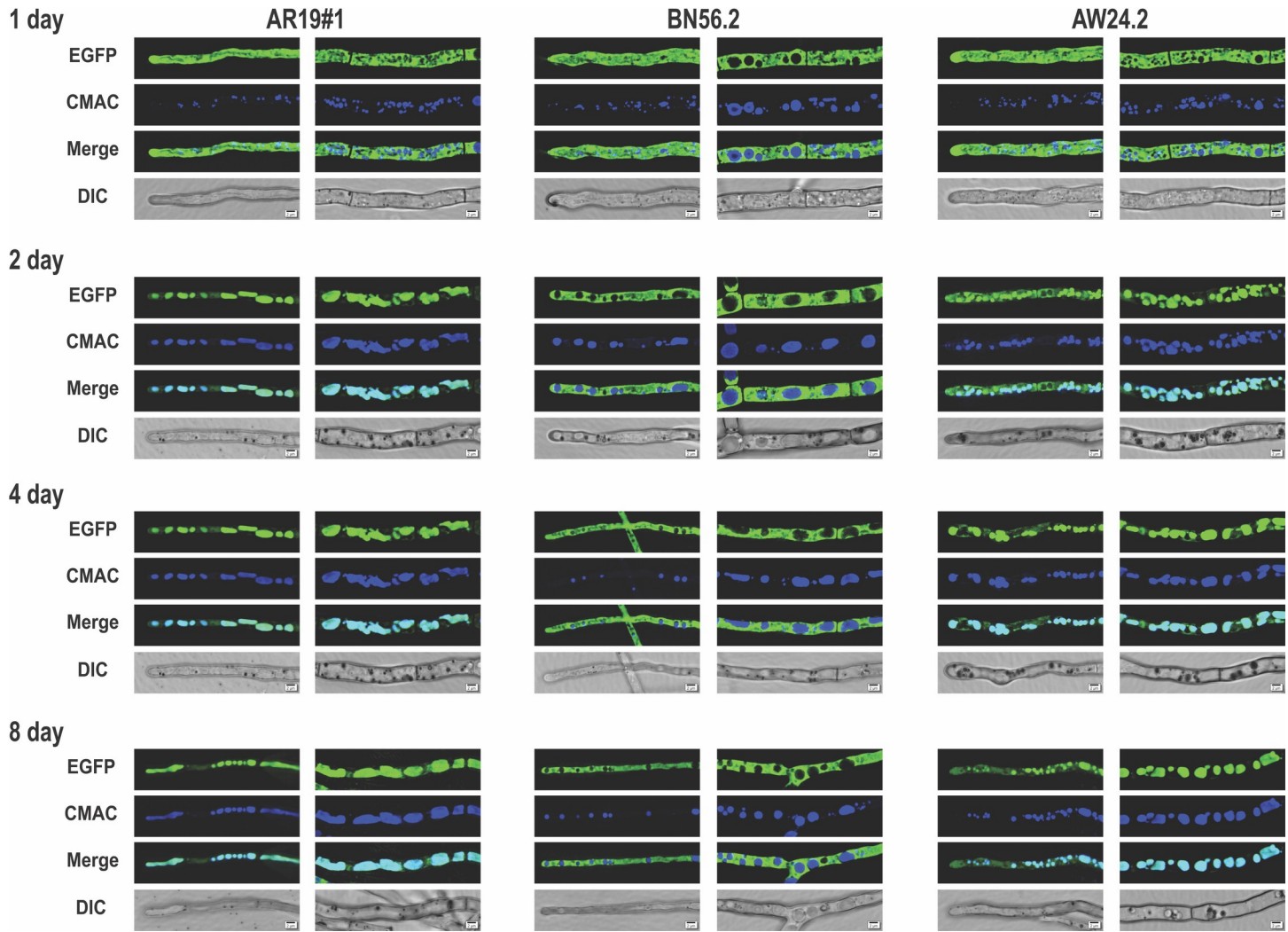

**Fig 3.** *A. niger* **mycelial morphology during acidogenic growth.** The morphology and EGFP fluorescence of *A. niger* strains, AR19#1 (wild type), BN56.2 (*Δatg1*) and AW24.2 (*atg1* complemented) was recorded on fermentation medium (AM) for 8 days. In each panel, the left frame shows the image of a tip cell and the right frame is of an intermediate cell. The mycelia were stained with CMAC when vacuoles appear blue; the vacuoles containing EGFP appear cyan on merge (scale bar = 2 μm).

MM ([Fig 4A], bottom panel) and none of them produced citrate when grown on MM. The citrate yield (gram citric acid produced per gram mycelial dry weight) on AM, at each time point was also calculated. On both counts (on the basis of citrate levels in the spent media and yield per mycelial dry weights) it is evident that disrupting autophagy negatively affects acidogenesis. These results clearly implicate an autophagy-like state in the acidogenic metabolism of *A. niger* and that a block in autophagy pathway negatively impacts citric acid production. This is in contrast to enhanced penicillin yields in *Δatg1* strain of *P. citrinum* [29] and increased ethanol production in *Δatg32* strain of yeast [46]. Obviously, we do not yet fully understand the mechanism(s) responsible for the unique acidogenic metabolism in *A. niger* [5,47,48].

## Conclusions

We have observed autophagy in response to nitrogen starvation and also by metabolically blocking ammonium assimilation in this fungus. Many phenotypic overlaps were noted

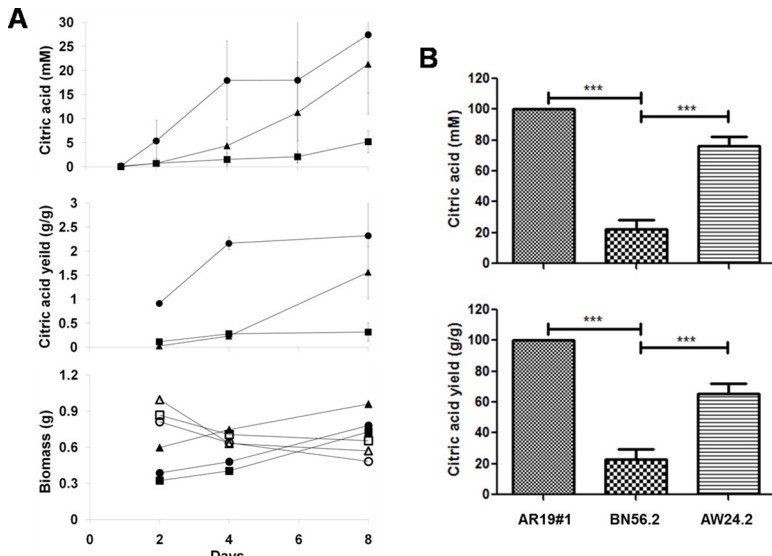

**Fig 4. Citric acid production by *A. niger*. (A)** The *A. niger* strains AR19#1 (•), BN56.2 (■) and AW24.2 (▲) were grown on fermentation medium (AM) and the citric acid in the spent medium (mM) and citric acid yield (per gram dry mycelia mass) are shown. Mycelial dry weights were measured for the strains AR19#1 (•), BN56.2 (■) and AW24.2 (▲) grown on AM and compared with corresponding data for growth on MM (respective open symbols; bottom graph). The error bars represent the data spread of five independent experiments. **(B)** Relative citric acid values (in percent) calculated from day 8 data for AR19#1, BN56.2 and AW24.2 strains are shown (*** $p < 0.0001$ with n = 5).

between autophagy and acidogenesis (Table 1). One key feature being unusual C:N balance and nitrogen limited growth of the fungus during acidogenesis. An extended autophagy-like state was observed throughout the acidogenic stages (idiophase) of fermentation and a block in autophagy pathway negatively impacted citric acid production. Both autophagy and acidogenesis are compromised in the *Δatg1* and *Δatg8* strains of *A. niger*. To our knowledge, this is the first report on the association of autophagy with the acidogenic metabolism of *A. niger*. Despite the extensive accumulated literature on the biochemistry and physiology of this fungus, a clear physiological cause-effect relation describing acidogenesis has not emerged so far [5,48]. There are various possibilities for how autophagy may perhaps support acidogenesis in *A. niger*. For instance, autophagy may—a) provide important metabolites to sustain acidogenesis, b) the enzymes (and the pathway) for citrate formation may possibly operate from the vacuolar compartment during acidogenesis, or c) simply keep the organism alive for citrate production to proceed. While how the phenomena of acidogenesis and autophagy are linked needs further study, underlying mechanisms could assist in improving citric acid fermentation.

## Supporting information

**S1 Fig. Effect of nitrogen starvation on the protein content of *A. niger* mycelia.** The total protein from the mycelial extracts (2 g wet weight; collected 24 h after transfer to either MM or MM-N) of *A. niger* strains AR19#1 (wild type), BN56.2 (*Δatg1*), AW24.2 (*atg1* complemented) and C6JT2 is shown (average of two separate experiments).
(TIF)

**S2 Fig. Effect of DMIP treatment on *A. niger* mycelial morphology.** The mycelia of *A. niger* strains AR19#1 (wild type) and BN56.2 (*Δatg1*) expressing EGFP in the cytoplasm were grown on minimal medium (MM) till 20 h and then transferred to MM+ DMIP, MM+acetone

(solvent control) and MM alone. The EGFP fluorescence of mycelia recorded after 6 h of DMIP treatment is shown. A total of 50 micrograph of each cell type were imaged (scale bar = 2 μm).
(TIF)

**S3 Fig. *A. niger* mycelial morphology during acidogenic growth.** The morphology and EGFP fluorescence of *A. niger* strains, AR19#1 (wild type), BN57.1 (*Δatg8*) and AW25.1 (*atg8* complemented) was recorded on fermentation medium (AM) for 8 days. In each panel, the left frame shows the image of a tip cell and the right frame is of an intermediate cell. The mycelia were stained with CMAC when vacuoles appear blue; the vacuoles containing EGFP appear cyan on merge (scale bar = 2 μm).
(TIF)

**S4 Fig. Citric acid production by *A. niger*.** The *A. niger* strains C6JT2 (○), AR19#1 (•), BN57.1 (■) and AW25.1 (▲) were grown on fermentation medium (AM) and the citric acid measured in the spent medium is shown.
(TIF)

**S1 Table. Primers used in this work.**
(DOC)

## Acknowledgments

The data related to *A. niger Δatg8* strain was recently generated and provided by Dr. Ejaj Pathan. We gratefully acknowledge him for the same.

## Author Contributions

**Conceptualization:** Baljinder Kaur, Narayan S. Punekar.

**Data curation:** Baljinder Kaur, Narayan S. Punekar.

**Formal analysis:** Baljinder Kaur, Narayan S. Punekar.

**Funding acquisition:** Narayan S. Punekar.

**Investigation:** Baljinder Kaur, Narayan S. Punekar.

**Methodology:** Baljinder Kaur, Narayan S. Punekar.

**Project administration:** Narayan S. Punekar.

**Resources:** Narayan S. Punekar.

**Supervision:** Narayan S. Punekar.

**Validation:** Baljinder Kaur, Narayan S. Punekar.

**Visualization:** Baljinder Kaur.

**Writing – original draft:** Baljinder Kaur, Narayan S. Punekar.

**Writing – review & editing:** Narayan S. Punekar.

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
