## [Decision Letter · Decision Letter 0]

29 Aug 2019

[EXSCINDED]

PONE-D-19-21095

Autophagy is important to the acidogenic metabolism of Aspergillus niger

PLOS ONE

Dear Dr. Punekar,

Thank you for submitting your manuscript to PLOS ONE. After careful consideration, we feel that it has merit but does not fully meet PLOS ONE’s publication criteria as it currently stands. Therefore, we invite you to submit a revised version of the manuscript that addresses the points raised during the review process.

Please address all of the reviewer comments, most of which are focused upon changes to the manuscript text. Please also pay special attention to the comments of Reviewer 1 regarding a potential revisitation of results from a previous paper.

We would appreciate receiving your revised manuscript by Oct 13 2019 11:59PM. To enhance the reproducibility of your results, we recommend that if applicable you deposit your laboratory protocols in protocols.io, where a protocol can be assigned its own identifier (DOI) such that it can be cited independently in the future. For instructions see: http://journals.plos.org/plosone/s/submission-guidelines#loc-laboratory-protocols

We look forward to receiving your revised manuscript.

Kind regards,

Cory D. Dunn, Ph.D.

Academic Editor

PLOS ONE

Journal Requirements:

2. Our internal editors have looked over your manuscript and determined that it is within the scope of our Autophagy and Proteostasis Call for Papers. This collection of papers is headed by a team of Guest Editors: Sharon Tooze, Fulvio Regiori and Thorsten Hoope. The Collection will encompass a diverse range of research articles from early initiation of autophagy, to understand the role other proteostasis pathways play in maintaining cellular homeostasis and the cross talk between the two. 

Additional information can be found on our announcement page: https://collections.plos.org/s/autophagy-proteostasis..

If you would like your manuscript to be considered for this collection, please let us know in your cover letter and we will ensure that your paper is treated as if you were responding to this call. If you would prefer to remove your manuscript from collection consideration, please specify this in the cover letter.

Reviewers' comments:

Reviewer's Responses to Questions

**Comments to the Author**

1. Is the manuscript technically sound, and do the data support the conclusions?

Reviewer #1: Yes

Reviewer #2: Yes

2. Has the statistical analysis been performed appropriately and rigorously? 

Reviewer #1: No

Reviewer #2: Yes

3. Have the authors made all data underlying the findings in their manuscript fully available?

Reviewer #1: Yes

Reviewer #2: Yes

4. Is the manuscript presented in an intelligible fashion and written in standard English?

Reviewer #1: Yes

Reviewer #2: Yes

5. Review Comments to the Author

Reviewer #1: In this manuscript, the authors seek to determine whether citrate production (acidogenesis) in Aspergillus niger requires autophagy. They do provide evidence that mutants compromised in autophagy have reduced citrate production, with the best evidence coming from reduced citrate production in the atg1 mutant (Figure 4). However, there are a number of issues that should be addressed before publication.

1. Table 1, which compares aspects of acidogenesis and autophagy, is portrayed as a major motivation for the study, but contains items that do not support the hypothesis that the two processes are related. Specifically, the effects of acidogenesis on germination and conidiation appear to be the same as those seen in the absence of autophagy. This would argue against the overall correlation of the organism’s state in autophagy and acidogenesis and are not “similarities” as indicated in the Table’s title. These differences should be discussed further and/or removed from the table.

2. Reference 25 has already established that autophagy occurs with nitrogen limitation, demonstrated relocation of GFP to the vacuole under these conditions, and showed that the relocation is prevented in an atg1 mutant. The authors largely appear to be reproducing these results in Figures 1 and 2 and should distinguish their studies from those in Ref. 25 more clearly.

3. The data in Figure 4A are the most important and novel in the paper, as they demonstrate that acidogenesis is severely compromised in the atg1 mutant. However, the basis of the error bars is not clear. In addition, the atg1 strain complemented with the wild-type ATG1 gene shows a significant delay in citrate production that is masked in Figure 4B by using a later time point. This delay should be discussed.

4. There is no real support for the statement that because of the vacuolar morphology changes, citrate formation might occur in the vacuole (lines 320-322). This is misleading and should be removed.

Reviewer #2: This manuscript examines the role of autophagy in acidogenesis in A. niger. Based on the observations that both increased acidogenesis and autophagy share a number of similar characteristics such as activation by nitrogen starvation and metals deficiency, impaired proliferation, and enhanced protein degradation, the authors test the hypothesis that autophagy is required for citric acid fermentation in this organism. They investigate this relationship by tracking cytosolic eGFP movement to the vacuole as a marker for autophagy. In parallel, the authors also examine citric acid production, both concentration and yield, to quantify acidogenesis in wild type and autophagy deficient mutants. The authors demonstrate that autophagy does occur in A. niger, and show that loss of autophagy diminishes citric acid production. The experiments are well controlled and executed, and the results are conclusive.

While the authors’ overall conclusion that autophagy is required for citric acid production is useful with respect to improving industrial citric acid fermentation yields, the findings are not particularly revealing about the mechanism by which autophagy supports citric acid production, which the authors acknowledge. Identifying this mechanism is clearly beyond the scope of this study, and the authors should address the following points before publishing.

Major points

1. The authors’ conclusion on lines 200-201 that the movement of eGFP to the vacuole represents autophagy is premature without yet showing the requirement for Atg1 or other autophagy machinery. Prior to the Atg1 experiments in Fig 2, the authors should not refer to eGFP vacuole localization as autophagy in the text or titles. For example, a more appropriate title for the first section would be “nitrogen starvation induces eGFP localization to vacuoles in A. niger.” The same holds true for the title and section starting on line 220 about ammonia assimilation.

2. The section on biomass yield beginning on line 210 should include the data in figure or table form.

3. Starting on line 319 of the conclusion, the authors state that their results suggest that enzymes for citrate production are localized to the vacuole. This statement seems a bit premature, and there are numerous possibilities for how autophagy may support citric acid production. For example, it’s possible that autophagy provides important metabolites to maintain citrate production and/or just keeps the organism alive so citrate production can proceed. The authors should modify this statement to include other possible interpretations of their data.

Minor:

Figure 4A: Remove curve fitting for scatterplot, and use linear interpolation lines instead.

Figures 1-3: Differentiate tip intermediate cell images from basal cell images and add arrows to show example of vacuole in images.

Lines 249-259: Consider moving this section into the introduction.

6. PLOS authors have the option to publish the peer review history of their article (what does this mean?). If published, this will include your full peer review and any attached files.

Reviewer #1: No

Reviewer #2: No

---

## [Author Response · Author response to Decision Letter 0]

23 Sep 2019

Professor Cory D. Dunn, Ph.D.

Academic Editor

PLOS ONE September 23, 2019

Dear Professor Dunn, 

We thank you and the reviewers for raising relevant issues that helped improve the quality of this work and presentation. 

We would like our manuscript to be considered for the PLoS collection of “Autophagy and Proteostasis” Call for Papers. 

We have now revised the manuscript (PONE-D-19-21095) entitled “Autophagy is important to the acidogenic metabolism of Aspergillus niger”, addressing all concerns of the reviewers. Our responses to each point brought up by the academic editor and reviewer(s) are appended below. Since the first submission of this manuscript we have also obtained additional results related to acidogenic metabolism where autophagy was manipulated by agt8 deletion (BN57.1 strain) and its subsequent complementation (AW25.1 strain). We have now included this atg8 related data on cellular morphology (as Figure S3) and citric acid production (as Figure S4) that additionally supports the findings and adds value to this study. The manuscript is accordingly modified at appropriate places.

Reviewer #1: 

In this manuscript, the authors seek to determine whether citrate production (acidogenesis) in Aspergillus niger requires autophagy. They do provide evidence that mutants compromised in autophagy have reduced citrate production, with the best evidence coming from reduced citrate production in the atg1 mutant (Figure 4). However, there are a number of issues that should be addressed before publication.

We have now included the data on cellular morphology (as Figure S3) and citric acid production (as Figure S4) with Δatg8 strain of A. niger that additionally supports the findings and adds value to this study.

1. Table 1, which compares aspects of acidogenesis and autophagy, is portrayed as a major motivation for the study, but contains items that do not support the hypothesis that the two processes are related. Specifically, the effects of acidogenesis on germination and conidiation appear to be the same as those seen in the absence of autophagy. This would argue against the overall correlation of the organismʼs state in autophagy and acidogenesis and are not “similarities” as indicated in the Tableʼs title. These differences should be discussed further and/or removed from the table.

The point made by the reviewer is well taken. We have accordingly modified the Table 1 title and also modified the related text (lines 41-43). 

2. Reference 25 has already established that autophagy occurs with nitrogen limitation, demonstrated relocation of GFP to the vacuole under these conditions, and showed that the relocation is prevented in an atg1 mutant. The authors largely appear to be reproducing these results in Figures 1 and 2 and should distinguish their studies from those in Ref. 25 more clearly.

That autophagy occurs with carbon depletion in submerged A. niger cultures is well established; but their study with respect to nitrogen limitation was restricted to surface cultures on solid MM [25]. Also, the Δatg8 strain was shown to be more sensitive to nitrogen limitation than the Δatg1 strain, whereas both deletion strains were comparably affected by carbon limitation. Acidogenesis is possibly a nitrogen limited state of submerged growth. It was therefore of interest to first demonstrate that nitrogen starvation leads to autophagy in two different A. niger strains, grown in liquid culture. This justification is now included (lines 202-208).

3. The data in Figure 4A are the most important and novel in the paper, as they demonstrate that acidogenesis is severely compromised in the atg1 mutant. However, the basis of the error bars is not clear. In addition, the atg1 strain complemented with the wild-type ATG1 gene shows a significant delay in citrate production that is masked in Figure 4B by using a later time point. This delay should be discussed.

The basis of error bars is now distinctly mentioned in the Figure 4 legend (line 376). The atg1 complementation was achieved by random integration (and was not by homologous recombination, Ref 25). This could possibly account for the observed delay in acid production. This possibility is clearly stated now (lines 353-357). The atg8 data freshly included as Figures S3 and S4 further corroborates the novelty of this work. 

4. There is no real support for the statement that because of the vacuolar morphology changes, citrate formation might occur in the vacuole (lines 320-322). This is misleading and should be removed.

While addressing the mechanism(s) that connect acidogenesis and autophagy are beyond the scope of this study, we wish to provide few possibilities for future study. As suggested by the Reviewer 2, we have now modified the text and included other possible interpretations for this linkage (lines 390-395). 

Reviewer #2: 

This manuscript examines the role of autophagy in acidogenesis in A. niger. Based on the observations that both increased acidogenesis and autophagy share a number of similar characteristics such as activation by nitrogen starvation and metals deficiency, impaired proliferation, and enhanced protein degradation, the authors test the hypothesis that autophagy is required for citric acid fermentation in this organism. They investigate this relationship by tracking cytosolic eGFP movement to the vacuole as a marker for autophagy. In parallel, the authors also examine citric acid production, both concentration and yield, to quantify acidogenesis in wild type and autophagy deficient mutants. The authors demonstrate that autophagy does occur in A. niger, and show that loss of autophagy diminishes citric acid production. The experiments are well controlled and executed, and the results are conclusive.

While the authorsʼ overall conclusion that autophagy is required for citric acid production is useful with respect to improving industrial citric acid fermentation yields, the findings are not particularly revealing about the mechanism by which autophagy supports citric acid production, which the authors acknowledge. Identifying this mechanism is clearly beyond the scope of this study, and the authors should address the following points before publishing.

Major points

1. The authorsʼ conclusion on lines 200-201 that the movement of eGFP to the vacuole represents autophagy is premature without yet showing the requirement for Atg1 or other autophagy machinery. Prior to the Atg1 experiments in Fig 2, the authors should not refer to eGFP vacuole localization as autophagy in the text or titles. For example, a more appropriate title for the first section would be “nitrogen starvation induces eGFP localization to vacuoles in A. niger.” The same holds true for the title and section starting on line 220 about ammonia assimilation.

The title of the first section is changed to “Nitrogen starvation induces EGFP localization to vacuoles in A. niger” (line 196). Also, the text is modified at line 218. To be consistent with this narrative the section “Blocking ammonia assimilation also induces autophagy in A. niger” is moved to a place after atg experiments. 

2. The section on biomass yield beginning on line 210 should include the data in figure or table form.

The biomass data is now included as Table 2 (lines 241-252) and is mentioned on line 235. The data on protein content (lines 235-237) is provided as a supplementary Figure S1.

3. Starting on line 319 of the conclusion, the authors state that their results suggest that enzymes for citrate production are localized to the vacuole. This statement seems a bit premature, and there are numerous possibilities for how autophagy may support citric acid production. For example, itʼs possible that autophagy provides important metabolites to maintain citrate production and/or just keeps the organism alive so citrate production can proceed. The authors should modify this statement to include other possible interpretations of their data.

While addressing the mechanism(s) that connect acidogenesis and autophagy are beyond the scope of this study, we wish to provide few possibilities for future study. As suggested by this reviewer, we have now modified the text and included other possible interpretations for this linkage (lines 390-395). 

Minor:

Figure 4A: Remove curve fitting for scatterplot, and use linear interpolation lines instead.

This is done and fresh Figure 4 is uploaded.

Figures 1-3: Differentiate tip intermediate cell images from basal cell images and add arrows to show example of vacuole in images.

The legends to these three figures are re-worded to include this information (and color codes described). 

Lines 249-259: Consider moving this section into the introduction.

We believe that this preamble (lines 307-317) is appropriate to the data being presented in the subsequent paragraphs. Since we do have a combined Results and Discussion section, we think it is best to retain the section as is.

---

## [Decision Letter · Decision Letter 1]

2 Oct 2019

Autophagy is important to the acidogenic metabolism of Aspergillus niger

PONE-D-19-21095R1

Dear Dr. Punekar,

We are pleased to inform you that your manuscript has been judged scientifically suitable for publication and will be formally accepted for publication once it complies with all outstanding technical requirements.

With kind regards,

Cory D. Dunn, Ph.D.

Academic Editor

PLOS ONE

Reviewers' comments:

Reviewer's Responses to Questions

**Comments to the Author**

1. If the authors have adequately addressed your comments raised in a previous round of review and you feel that this manuscript is now acceptable for publication, you may indicate that here to bypass the “Comments to the Author” section, enter your conflict of interest statement in the “Confidential to Editor” section, and submit your "Accept" recommendation.

Reviewer #1: All comments have been addressed

Reviewer #2: All comments have been addressed

2. Is the manuscript technically sound, and do the data support the conclusions?

Reviewer #1: Yes

Reviewer #2: Yes

3. Has the statistical analysis been performed appropriately and rigorously? 

Reviewer #1: Yes

Reviewer #2: Yes

4. Have the authors made all data underlying the findings in their manuscript fully available?

Reviewer #1: Yes

Reviewer #2: Yes

5. Is the manuscript presented in an intelligible fashion and written in standard English?

Reviewer #1: Yes

Reviewer #2: Yes

6. Review Comments to the Author

Reviewer #1: The manuscript is now acceptable. The authors have satisfied the points raised in my previous review.

Reviewer #2: (No Response)

7. PLOS authors have the option to publish the peer review history of their article (what does this mean?). If published, this will include your full peer review and any attached files.

Reviewer #1: No

Reviewer #2: No

---

## [Editor Report · Acceptance letter]

4 Oct 2019

PONE-D-19-21095R1 

Autophagy is important to the acidogenic metabolism of Aspergillus niger 

Dear Dr. Punekar:

I am pleased to inform you that your manuscript has been deemed suitable for publication in PLOS ONE. Congratulations! Your manuscript is now with our production department. 

With kind regards,

on behalf of

Dr. Cory D. Dunn 

Academic Editor

PLOS ONE